# Applications of Probiotic Constituents in Cosmetics

**DOI:** 10.3390/molecules28196765

**Published:** 2023-09-22

**Authors:** Jiaxin Dou, Ning Feng, Fangyu Guo, Zouquan Chen, Jie Liang, Ting Wang, Xueping Guo, Zhenshang Xu

**Affiliations:** 1State Key Laboratory of Biobased Material and Green Papermaking, Qilu University of Technology, Shandong Academy of Science, Jinan 250353, China; 17864099662@163.com (J.D.); gfangyu2022@163.com (F.G.); 18827174212@163.com (Z.C.); liangjie122610@163.com (J.L.); 2School of Bioengineering, Qilu University of Technology, Shandong Academy of Science, Jinan 250353, China; 3Huaxi Biotechnology Co., Ltd., No. 678, Tianchen Road, Lixia District, Jinan 250000, China; jiefeng93@163.com; 4Zhucheng Dongxiao Biotechnology Co., Ltd., Zhucheng 262200, China

**Keywords:** probiotics, beneficial microorganisms, cosmetics, anti-aging, dermatology, skin care

## Abstract

Over the past few decades, research on the benefits of beneficial microorganisms on skin health has expanded and attracted a lot of attention. Today, a wide range of probiotic products are becoming available. With their extensive component profiles and varied physiological effects, probiotics, as well as extracts of them, have a significant impact on cosmetics. However, the present boom in consumer interest in alternatives has broadened the probiotic industry’s research and development frontiers. Considering the foregoing, it should come as no surprise that probiotics are highly valued for their proven anti-aging, skin whitening, anti-inflammatory, and photoprotective effects. This review aims to compile information on probiotics’ properties, their extracts, and preparations used in cosmetics. It also further summarizes research and applications on probiotic fermentation to promote the use of probiotic fermentation products in cosmetics. Notably, this review also adds information on particular properties and mechanisms of action of probiotics, which fills a gap in the research and application of probiotics in skin treatment and care. Their antioxidant and anti-aging qualities have received particular consideration. This review provides a new basis for the broad application of probiotics in cosmetics.

## 1. Introduction

Cosmetics are described by the US Food and Drug Administration (FDA) as “products (excluding pure soap) used to cleanse, beautify, enhance attractiveness, or alter the appearance of the human body”. Products that care for skin, hair, and mouth meet this standard. Probiotics and postbiotics are two product categories connected to the microbiome developed by the International Cosmetic Regulatory (ICCR) Collaboration [1]. Probiotic is “a living microbe that, when administered in sufficient amounts, provides a health benefit to the host” [2]. Probiotic products must meet three essential criteria. (1) To use the strain for its intended purpose, it must be genetically and phenotypically described and supported by results from experiments published in peer-reviewed journals. (2) At the time of use, the product must have a quantity of live microorganisms similar to the product demonstrated in clinical studies to benefit the designated target site. (3) If people are the intended receivers, the delivery technique, dosage, and length of use should be determined through human studies [3,4]. The microbiological content of cosmetics is anticipated to be modest (less than 500 Colony-Forming Units/gram (CFUs/g) for eye products and less than 1000 CFUs/g for other products) for safety reasons. There can be no natural probiotic cosmetics if it is impossible to include live bacteria. They may still have advantageous components from probiotic strains, in any case. Metazoons, also called bacterial lysates, ferments, and filtrates, are “preparations of non-living microorganisms or their components that have health benefits for the target host [5].” They can also be applied to the skin and mucous membranes of the host. These criteria do not cover purified metabolites or components that are cell-free. Filtrates are not considered epigenetic if they do not have biological components. Bacterial lysates and ferments may fit into this category depending on their makeup. Safety, functionality, and technical utility have been identified as essential criteria for the selection of probiotic bacteria, according to the World Health Organization (WHO), the Food and Drug Administration (FDA), and the European Food Safety Authority (EFSA). Experiments can be used to determine if microbial strains meet the requirements mentioned above in vitro. It is possible to screen microorganisms for their potential as probiotic strains based on these selection criteria confirmed by in vitro experiments [1].

Figure 1 illustrates the main processes by which probiotics promote skin health. Probiotics have been demonstrated to possess a number of skin-beneficial features, including the ability to reduce skin inflammation, heal several skin conditions, and shield against allergic contact dermatitis. Additionally, they are essential for enhancing the skin barrier, promoting water absorption, and delaying the aging process of the skin [6]. These data offer a theoretical foundation for creating probiotic products.

More and more studies have proven that probiotics have many benefits for the skin. Figure 2 shows the effects of probiotics in terms of anti-aging, antioxidants, whitening, and anti-wrinkling, which have increased the addition of probiotics in cosmetics.

Probiotics have recently been used in common care items. These cosmetics, such as face creams, moisturizing lotions, tonics, body washes, hair products, and beauty masks, contain pieces of cell walls and inert bacteria. Since live bacteria cannot be found in cosmetics, several companies make probiotic skin care products and add some probiotics. Some people will include prebiotics (probiotic “food,” such as oligosaccharides, galactose oligosaccharides, and fructooligosaccharides), which can prevent the growth of dangerous bacteria from readjusting the makeup of the skin microbiome while supporting the growth of helpful bacteria [7]. Others will add bacteriocin (the “active products” of probiotics, metabolites, such as lactobacillus fermentation products and yeast fermentation product extracts) to skin care products in order to assist the skin in readjusting the micro-ecological balance because of its small molecules, good stability, high-temperature resistance, acid and alkaline opposition, and acid and alkaline resistance. Probiotic extracts are now utilized increasingly frequently in cosmetics; lactobacillus is the most common ingredient. Various goods are available because of the abundance of producers and cosmetic forms, and it is challenging to calculate their precise quantity because new items enter the market virtually every year. Table 1 provides examples of 35 cosmetic products using probiotics or probiotic fermentation broth.

Probiotics themselves and their extracts play an essential role in the cosmetic market. There is an increasing number of cosmetics containing probiotics, especially those using probiotic fermentation products. Therefore, there is a need for authors to provide a review to focus on probiotics in cosmetics. Currently, old review articles in the field do not detail the mechanism of action of probiotics in cosmetic applications. The present review describes, in detail, the two significant skin challenges, anti-aging and antioxidant, from in vitro and ex vivo experiments to the mechanism of action. Below are a few older review articles that have been published in this area. Jinyan Yu et al. [8] reviewed the effects of probiotics on skin whitening, moisturizing, anti-aging, improving skin wrinkles, and deodorizing, which provided a new rationale for the broad application of probiotics in skin care. The review by Scarlett Puebla-Barragan et al. [4] discussed the current market, regulatory aspects, and potential applications of probiotics in the personal care industry. Ting Gao et al. [9] reviewed the application and mechanism of probiotic-mediated gut microbiota in skin care, providing a new rationale for the wide application of probiotics in skin care. Marco Duarte et al. [10] conducted a study on some postbiotics derived from probiotics. The review focuses on what is currently known about these compounds, the benefits of their use, the main postbiotics products available on the market and the players, the main trends in production, and the production methods available. Arun Karnwal et al. [11] reviewed the application of microbial biosurfactants instead of chemical surfactants in existing cosmetic and personal skin care pharmaceutical formulations. Some of the previously published references mentioned above also mention probiotics’ antioxidant and anti-aging properties. However, concerning skincare rather than cosmetic applications, they describe microbial biosurfactants and postbiotics rather than probiotics and their extracts. The present review aims to collect information about probiotics and their extracts and formulations for cosmetic applications.

Twenty years ago, the terms probiotics, prebiotics, and microbiome were unknown in the cosmetics industry. Although chemistry is a foundational element of the cosmetics industry, it has not yet been properly applied to molecules that identify the advantages of microbial products. Given the growth of the microbiome area, it is essential that professionals with knowledge of microbiology and chemistry arise in order to guarantee that consumers receive high-quality cosmetics that adhere to the definitions of probiotics, prebiotics, etc. There is no question that the inclusion of probiotics in cosmetics could result in novel approaches to enhancing both look and health, which would provide regulatory difficulties for the entry of cosmetics into the health sector. While promoting regulatory improvements, we must insist on clinical validation, verification of product safety, and strict guidelines for handling, storing, and using items that contain microbes and their byproducts or cell walls. Unfounded assertions are useless to everyone, while scientific research yields goods that are very beneficial to human health and well-being.

## 2. Application of Probiotics in Anti-Aging Cosmetics

The two types of skin aging are natural aging and photoaging. The term “natural aging” describes the contribution of internal components in the body brought on by exposed and unexposed areas, most notably manifested by wrinkles and skin relaxation. Ultraviolet radiation can cause photoaging when it enters the skin, which is a significant contributor to aging skin changes and cancer of the skin [12]. Changes in skin-related microbial communities, elevated skin pH, aberrant reactive oxygen species (ROS) generation, decreased collagen levels, and altered immune response are characteristics of photoaged skin.

The current probiotic cosmetics offer anti-aging effects on skin tissues primarily through the following mechanisms, per studies on the physiology of skin and aging mechanisms: hydrating and mending skin barrier function, replenishing collagen and elastin in the skin, and antioxidants [13]. In Table 2, various research investigations on probiotics’ antioxidant and anti-aging effects are compiled.

With the intensification of the aging of society and the pursuit of youth, the share of anti-aging skin care products in the cosmetics market has gradually increased. Consumers’ demand for the safety and efficacy of anti-aging cosmetics has also promoted the continuous improvement and improvement of the research and development technology of anti-aging cosmetics.

### 2.1. Hydrate and Restore the Skin’s Barrier

Moisturizing and repairing the skin’s barrier function are two crucial steps anti-aging cosmetics must take to address the issue of aging skin. Dry skin, a crucial sign of aging, will result from a reduced epidermis barrier function. Natural moisturizing components in keratinocytes and sebum membranes, which shield the skin’s surface and keep water from evaporating, keep the skin moisturized. Deteriorated sebaceous membranes, impaired epidermal barriers, and finally drier skin are all symptoms of aging skin. In order to make anti-aging cosmetics more effective, substances with moisturizing and skin barrier repair properties might be incorporated. For instance, adding the bioactive substances *Streptococcus thermophilus* S244 and *Streptococcus salivarium* spp. to cosmetic formulae can result in the production of enzymes that hydrate the skin, reducing dryness and slowing the aging process [20]. Figure 3 displays the moisturizing and anti-aging properties of probiotics.

According to clinical trials, oral probiotics can enhance skin hydration and increase skin water content, which can reduce wrinkles. In hairless mouse epidermis, oral treatment of *Bifidobacterium breve* or *Lactobacillus plantarum* decreased UV-induced water loss [21,22]. It is possible to increase skin water content and decrease wrinkles by taking a *L. plantarum* lysate or heat-inactivated lactococcus lactis orally [24,25]. On this basis, Hiroko Nakai et al. [27] studied the local application of heat-inactivated *Lactobacillus plantarum* L-137 (HK L-137). They found that treating epidermal cells with the cell fluid of HK L-137 (preparation of HK L-137 [38]) could increase the content of hyaluronic acid (HA). The mechanism of its action is that HK L-137 induces the production of cytokine IFN-γ to activate the NFκB pathway through a synergistic effect with TNF-α to enhance the expression of hyaluronate synthase mRNA in fibroblasts, thus producing more HA to play a role in moisturizing and anti-wrinkling.

The barrier and water retention functions of a healthy stratum corneum depend heavily on ceramide. According to studies, *Streptococcus thermophilus* preparations with ultrasonic therapy can boost the amount of ceramide in the stratum corneum, enhancing the barrier function and keeping the stratum corneum flexible [39].

### 2.2. Antioxidation

The excessive attack of free radicals on biological tissues like chromosomes, mitochondria, cell membranes, and connective tissues would result in the body aging, claims the free radical theory of aging [18]. Unsaturated fatty acids in the skin will be impacted by high quantities of free radicals, leading to unstable lipid peroxidation that will eventually break down into malondialdehyde. Malondialdehyde will quickly attack proteins and phospholipids and react to create lipofuscin, a lipid–protein complex that builds up in cells and is a marker of cell aging. As a result, eliminating many ROS is essential for preventing aging.

The information indicates that superoxide dismutase (SOD), glutathione (GSH), vitamin E (V_E_), and coenzyme Q_10_ (Q_10_) are the primary active raw materials with antioxidant action. A biological antioxidant enzyme called SOD efficiently eliminates too many superoxide free radicals from the body. The primary function of V_E_ in the body is as an antioxidant; on the one hand, it can eliminate free reactive oxygen radicals and stop the production of lipid peroxide. On the other hand, it can safeguard SOD and lessen the pace at which soluble collagen becomes insoluble. Similar to V_E_, coenzyme Q_10_ is a naturally occurring antioxidant made by the cell itself and one of the elements that make up the mitochondrial respiratory chain.

We are aware that ROS generated by UV irradiation can increase the expression level of matrix metalloproteinases (MMPs), leading to the induction of apoptosis. ROS are also known to contribute to skin aging and pigmentation. Experimental proof of the antioxidant activity of heat-killed *Lactobacillus acidophilus* KCCM12625P (AL) was provided by Hye Yeon Lim et al. [40]. In human keratinocytes and human dermal fibroblasts (HDFs) exposed to UV light, AL was discovered to regulate the levels of ROS and MMPs. AL is, therefore, a cosmetic component that has anti-aging properties. AL can also dilute UV-induced melanin and reduce UV-induced pigmentation in B16F10 murine melanoma cells at the same time. Additionally, research has shown that heat-inactivated *Lactobacillus acidophilus* can reverse UV-induced skin damage [41] and boost the antioxidant defenses of the skin [17], which is therapeutically employed to prevent wrinkle formation.

Through an experimental study, Xiaofang Zhang et al. [16] discovered the anti-wrinkle effect of heat-killed *Lactobacillus rhamnosus* (RL). After receiving RL, mouse skin fibroblasts’ antioxidant capacity was increased and their ROS content was decreased, and human epidermal melanocytes were also able to produce anti-melanin, which might be used as an anti-photoaging ingredient in cosmetics.

In the realm of antioxidant research, where the combination of probiotics and antioxidants has synergistic effects on one another, β-Nicotinamide Mononucleotide (NMN) and lactic acid bacteria (LAB) have been investigated more and more. The combination of NMN and *Lactobacillus fermentum* TKSN041 has been shown in studies to be able to increase the levels of SOD, catalase (CAT), and interleukin in the skin, reduce UV-induced oxidative damage to the skin, and improve the overall antioxidant capacity of the skin. It is anticipated to become a potent drug for the prevention and treatment of skin photoaging [42].

Probiotics can create metabolites such as probiotic peptides, organic acids, flavonoids, alcohols, and polyphenols after heating some bacterial components such as lipoteichoic acid, peptidoglycan, exopolysaccharides (EPS), and cell surface proteins. Human dermal fibroblasts’ oxidative damage and photoaging have been shown to be reduced by lipoteichoic acid, which is extracted from probiotics [15]. In normal dermal fibroblasts and mouse melanoma cells, heat-killed *Lacticaseibacillus paracasei* (PL) can reduce UV-B-related oxidative damage and photoaging [14]. MMP expression, as well as extracellular matrix-degrading enzymes, were enhanced by UVR irradiation. MMP-1 is an interstitial collagenase that breaks down collagen’s triple helix among the MMP family of enzymes. We discovered that MMP-1 expression was reduced by lipoteichoic acid that was extracted from the cell wall of *L. plantarum*. Additionally, it can encourage collagen synthesis and hinder its breakdown, which both aid in lowering ROS production [43]. The theoretical underpinnings for the creation of active ingredients in skin antioxidants and anti-photoaging treatments are provided by this evidence.

### 2.3. Other Anti-Aging Effects

Probiotics include bioactive compounds in their extracellular vesicles, which are beneficial for skin health and anti-aging. Lipopolysaccharides, for instance, are found in the extracellular vesicles of gram-negative bacteria [26], while lipoteichoic acid is found in the extracellular vesicles of gram-positive bacteria [44]. These chemicals have a wide range of response mechanisms at their disposal. According to Chan Song Jo et al.’s research [45], women in their 20s had an average of more *L. plantarum* in their skin than women in their 50s. Extracellular vesicles (EVs) secreted by *L. plantarum* were found to have a number of anti-aging properties. These characteristics include their ability to prevent the activity of MMP-1 and elastase, enhance skin elasticity, and preserve cell shape. Second, they raises the level of filaggrin mRNA expression, which is necessary for epidermal homeostasis and the preservation of skin barrier function. With age, the expression of filaggrin is usually decreased [28], which increases the composition of the extracellular matrix, which is conducive to maintaining cell morphology.

Additionally, studies in humans have shown that EVs can lessen photoaging and age-related pigmentation of eye wrinkles. EVs are thus a potent anti-aging ingredient for skin treatments. By concurrently inhibiting melanin development and enhancing UV-A absorption with the help of *L. plantarum* SM4 biotransformation, the inner shell of the chestnut can be used as a cosmetic material [46].

## 3. Using Probiotics to Whiten Skin and Get Rid of Freckles

The amount of melanin produced by pigmented cells in the epidermis and its distribution in the top layer of skin, where melanin is transferred and amplified by prolonged exposure to sunlight or ultraviolet radiation, determines the color of human skin.

The need for whitening goods among young people is rising in modern society. There are several diets, medications, and cosmetics on the market today that have whitening properties. Arbutin, kojic acid, and niacinamide are a few melanin-producing inhibitors that have recently been discovered and utilized as skin lighteners [47,48,49]. However, the usage of naturally occurring chemicals rather than chemically synthesized ones has drawn greater attention.

### 3.1. Tyrosinase Activity Inhibition

Numerous enzymes and chemical catalytic processes are involved in the formation of skin melanin. Tyrosinase is essential for the production of melanin. Tyrosinase activity can be inhibited to reduce melanin formation, lighten the skin, and produce the effect of whitening. Figure 4 shows the mechanism of action of tyrosinase inhibition by probiotics. Because of their low toxicity and high absorption, tyrosinase inhibitors generated by probiotics are regarded as promising potential skin whitening agents [50]. Recent years have seen a rise in the use of probiotics in whitening cosmetics, such as the lactic acid in LAB, which can directly reduce melanin formation by inhibiting tyrosine. They can also whiten skin by interfering with tyrosinase expression. Tyrosinase was likewise suppressed by the probiotic culture’s supernatant. The anti-tyrosinase effect was seen in the supernatant of *Bifidobacterium* IDCC 4201 and *L. plantarum* IDCC 3501, which decreased melanin synthesis and altered the expression of proteins involved in the melanin formation pathway. A potential tyrosinase inhibitor is phenylacetic acid’s metabolite. The generation of phenylacetic acid’s anti-melanin may be boosted by the synergistic interaction of other functional molecules in the supernatant of the probiotic culture medium. Consequently, a probiotic culture medium supernatant with a high phenyl lactic acid content has the potential to be used in food and medicine as an anti-melanogenic agent [29].

The meaning of the abbreviations in the chart is as follows: adrenocorticotropic hormone (ATH), a-melanocyte activator (a-MSH), basic fibroblast growth factor (BFGF), cyclic adenosine monophosphate (cAMP), endothelin vascular peptide-1 (ET-1), granulocyte-macrophage-activating factor (GM-SCF), melanin receptor-1 (MCR-1), prostatic enzyme E2 (PGE2), protein kinase A (PKA), protein kinase C (PKC), phospholipase C (PLC), stem cell factor (SCF), and adenylate cyclase (ATC).

There are currently chemicals used in cosmetics and pharmaceutical products that can inhibit the melanin synthetase tyrosinase in melanocytes. However, the majority of the compounds that are now on the market can only stop the production of melanin; they cannot get rid of melanin that has already been created and implanted. They are less effective at lowering the amount of melanin in the skin. Kimchi-derived *Staphylococcus lactis* PMC48 can destroy melanin, degrade melanin, and inhibit tyrosinase, according to Sukyung Kim et al. It is superior to current melanin production inhibition approaches and is anticipated to be highly valuable as a raw material for cosmetics and medications that break down melanin [30].

### 3.2. The Role of Lipoteichoic Acid

Lipoteichoic acid is the main component of the cell wall of gram-positive bacteria. The modulation of the gastrointestinal and immunological systems, skin moisturization, and photoaging are only a few of the many positive effects of lipoteichoic acid, which is obtained from *L. plantarum*. Lipoteichoic acid’s use in the management of pigmentation was investigated by Hye Rim Kim et al. By activating the ERK and PI3K/AKT pathways, it reduced tyrosinase activity in cells and hematopoietic enzyme expression. Lipoteichoic acid, which was discovered from *L. plantarum*, can, therefore, treat melanosis and be employed as a cosmetic bleaching agent [31]. The formation of anti-melanin is aided by the lipoteichoic acid of *L. plantarum* GMNL6, which also functions as a cosmetic regulating substance [51].

### 3.3. Other Whitening Effects

The volunteers’ skin hydration, skin tone, spots, wrinkles, UV spots, and porphyrin syndrome were all improved after using a cream containing *L. plantarum* [32]. Staphylococcus sphaerus extract, an anti-melanin-producing agent, can reduce the pigmentation caused by melanocytostimulus [33]. These probiotics produce substances that are widely used in both medicine and cosmetics.

## 4. Probiotics Used in Anti-Inflammatory Cosmetics

In light of the rapid expansion of oral probiotics, numerous topical probiotic formulations have been proposed to treat skin micro-ecological diseases and encourage immunological homeostasis by balancing skin microbiota [52]. To maintain the skin’s micro-ecological balance, they produce antibacterial and anti-inflammatory substances to stop pathogen invasion and the growth of opportunistic microorganisms [35].

Topical probiotics dramatically reduced the signs and symptoms of rosacea, atopic dermatitis, and acne in a few small-scale clinical studies. Although the exact mechanism is uncertain, probiotics are thought to have anti-inflammatory effects via promoting regulatory T cells, producing anti-inflammatory cytokines (such IL-10), competing with pathogens for nutrients, and aggregating and replacing pathogens. Probiotic strains with symbiotic skin microorganisms, like *Lactobacillus*, *Bifidobacterium*, and *Streptococcus*, have been linked to skin immune-modulating effects by preventing the development of biofilms, lowering cytokines that cause systemic inflammation, and directly competitively inhibiting binding sites [53].

### 4.1. Cutaneous Inflammation

In their studies, Ia Khmaladze et al. [34] found that *Lactobacillus reuteri* DSM 17938 and its lysates might reduce skin irritation brought on by UVB (ultraviolet B, wavelength 280 nm~320 nm, mid-frequency medium-wave) exposure. Bacteria that are detrimental to the skin (*Staphylococcus aureus*, *Streptococcus pyogenes*, *Bacillus acnes*, and *Pseudomonas aeruginosa*) are inhibited by live *L. reuteri* DSM 17938. It is possible to cure inflammatory skin diseases using this strain. Heat-killed *Lactococcus lactis* H61 water extract is mixed with cosmetic ingredients for topical application. The inflammation of skin cells brought on by UVB loss was shown to be reduced by its ability to block the angiotensin-converting enzyme (ACE) (topical administration of angiotensin-converting enzyme inhibitors may enhance photoaged skin [54]). Interleukin-8 (IL-8) levels caused by UV radiation were lowered in cells pretreated with the extract, which also absorbed electromagnetic radiation in the UVB range [36]. This demonstrates that the extract has a number of advantages, such as anti-inflammation of the skin and UV damage prevention. The *Bifidobacterium longum* (BL) lysate possesses anti-inflammatory properties. Adding the BL lysate to cosmetics can improve various parameters related to inflammation (such as vasodilation, edema, mast cell degranulation, and reduced TNF-α release). The local use of BL preparation promotes skin homeostasis and guards against adverse environmental impacts that cause skin sensitivity (cold in winter, dry air) [37].

Atopic dermatitis (AD), also known as atopic eczema, is a chronic inflammatory skin condition that frequently flares up and is characterized by dry, itchy skin [55]. An essential component in the pathophysiology of atopic dermatitis is the microbial environment. Extracellular vesicles released by bacteria have been identified to affect allergic inflammatory processes, according to recent investigations. It was shown that the extracellular vesicles formed from *L. plantarum* were effective in avoiding skin inflammation by comparing the makeup of bacteria-released extracellular vesicles between atopic dermatitis sufferers and healthy subjects [56].

In patients with AD, *Staphylococcus* aureus predominates the skin microbiota, and bacterial burden is correlated with disease severity. *Staphylococcus aureus* colonization in patients with AD can be decreased using heat-treated cosmetic lotions containing *Lactobacillus johnsonii* NCC 533 (HT La1) [19].

The composition of the stratum corneum’s lipids has a crucial role in how well the osmotic barrier and water retention capacity work. Ceramide [23] is the primary component of them. Recent research has revealed that aberrant skin function in AD patients may result from a decrease in the total amount of ceramide. The increase in ceramide levels may be attributed to the hydrolysis of sphingomyelin-by-sphingomyelin enzymes present in the bacterial extract, which were used in an experimental cream to inactivate *S. thermophilus* using ultrasound. This finding supports the contribution of these bacterial lysates to stratum corneum barrier function. The experimental cream’s topical use also reduced the erythema, scales, and pruritus associated with AD skin [57].

### 4.2. Acne

Acne is a long-lasting inflammatory skin condition that affects sebaceous glands and hair follicles. Skin rashes, including whiteheads, blackheads, pustules, papules, and cysts on the face, chest, back, and shoulders are the primary clinical symptoms. The main contributory factors to acne vulgaris are androgen, excessive keratosis at the opening of hair follicles and sebaceous glands, bacterial colonization, and increased sebum output. In some patients, endocrine and genetic disorders may also play a role. Studies suggest that probiotics may reduce acne. The demand for probiotic supplements and beauty products is surging. Hope exists for acne sufferers’ skin using home remedies, like probiotic supplements and cosmetics.

Most acne cosmetics have the potential to improve clinical results. The cleanser should be used by all acne sufferers; those containing benzoyl peroxide or azelaic acid/salicylic acid/triclosan showed the most effectiveness. Products for managing sebum that contain zinc or niacinamide aid in reducing excessive sebum production. Acne healing could be sped up with the use of cosmetics containing antibacterial and anti-inflammatory substances such ethyl lactate, phytosphingosine, niacinamide, or resveratrol. Topical comedolytic drugs can help the skin absorb topical pharmaceuticals and have comedolytic effects. Examples include retinal/glycolic acid and lactic acid. Finally, a specific moisturizer should be seriously considered by all acne sufferers.

In order to get rid of potential skin pathogens, normal human skin can release a variety of antibacterial compounds. A gram-positive bacteria called *L. plantarum* makes antimicrobial peptides that can reduce inflammation and strengthen the skin’s natural defenses against microorganisms. We examined how *lactobacillus* extract affects the skin’s barrier function, the number of chemical irritants it produces, the skin microbiota, and acne-related erythema. The findings indicated that *lactobacillus* extract could successfully cure mild acne lesions and minimize skin erythema, repair the skin barrier, and diminish the skin microbial community [58].

The local ultrasonic treated preparation of *S. thermophilus* can raise the level of ceramide in the stratum corneum and assist in the treatment of acne because the ceramide (plant sphingosine) in *S. thermophilus* has antibacterial and anti-inflammatory effects against *Cutibacterium acnes* [59]. Topical applications of *Lactobacillus salivary* LS01 and *Bifidobacterium brevis* BR03 can also be used to treat rosacea [60]. Probiotics administered topically can act as a barrier to stop acne caused by other skin-invading pathogens [61]. Figure 5 demonstrates the role of the above probiotics in acne treatment. To combat harmful microbes and reduce inflammation, probiotics have lately been added to everyday skincare products.

## 5. Application of Probiotic Fermentation

Probiotic fermentation is a recent finding that can alter the chemical structure of medications to improve epidermal absorption and lower toxicity for the best possible pharmacological effectiveness. The aqueous extract of Acanthopanax Korean root (AE), which has undergone fermentation by *L. plantarum* and *Bifidobacterium bifidum*, can lessen intracellular ROS brought on by UVB or H_2_O_2_. Acanthopanax Korean root also exhibited much stronger antioxidant and anti-aging properties than extracts made prior to fermentation [62]. After *L. rhamnosus* fermentation, patchouli leaf extract can enhance the skin’s ability to fend off photoaging. According to studies, the fermented extract can increase the skin’s levels of SOD and GSH while lowering ROS [63]. Treatment with *L. rhamnosus* fermented jasmine extract can significantly slow down collagen breakdown, premature aging, and skin cell aging brought on by UVB and H_2_O_2_ [64]. It can also prevent excessive ROS production, collagen degradation, and premature aging in dermal fibroblasts. A possible therapeutic agent for the prevention and treatment of skin photoaging is *L. plantarum* fermented blackberry, which can diminish the production of wrinkles and help human dermal fibroblasts resist photoaging [65]. In human skin fibroblasts, UV-induced MMP-1 production and collagen degradation are prevented by fermenting lavender extract with *Pediococcus pentosaceus* DK1 [66]. The physiological activity of fructus aurantii extract is greatly enhanced by deep fermentation with *Lactobacillus brevis* [67]. The antioxidant activity of HaCaT cells can be increased by pomegranate peel and schisandra extracts that have been fermented by *Lactobacillus acidophilus* and *L. plantarum*, respectively [68]. *Lactobacillus helveticus* can ferment whey and decrease the expression of MITF, which prevents tyrosinase from blocking melanin synthesis [69]. The photoaging of fibroblasts brought on by UVA (ultraviolet A, wavelength 400 nm~320 nm, low-frequency long-wave) can be lessened by rhodiola rosea fermented by *L. plantarum* [70]. These fermented extracts contain anti-aging (Figure 6) and skin-lightening properties that could increase the usage of probiotics in the beauty industry. Probiotics’ fermented by-products serve a variety of purposes as well. The *Lactobacillus rhamnosus* LRH113 strain’s fermentation supernatant exhibits whitening and moisturizing properties, and it has a great deal of promise for use in future maintenance products.

## 6. Conclusions and Prospects

The use of probiotics and their extracts in cosmetics is discussed in this research, encompassing several application areas such as anti-aging, antioxidant, whitening, and anti-inflammatory effects. It summarizes the progress of research on probiotic fermentation. This review also collects new findings on probiotic research in dermatologic therapy and skin care. The beneficial function of probiotics on the skin has been further demonstrated with the development of analytical tools and biological activity evaluation methodologies. Since topical administration of probiotics has no known sensitizing or stimulating effects, there is still much potential for research in this area.

Based on the research progress collected in this review on probiotics in the field of cosmetics. In future research, it is crucial to undertake probiotic fermentation-type bioactivity investigations in order to learn more about the connection between the manufacturing process and the biological effects attained. On the other hand, additional studies of all probiotic strains on human skin or people are required to corroborate the therapeutic advantages shown in in vitro models.

Indeed, there should be a lot more research conducted in the future about adding probiotic ingredients to cosmetics, and some of the unexplored areas of the field will slowly come to the attention of researchers.

## Figures and Tables

**Figure 1 molecules-28-06765-f001:**
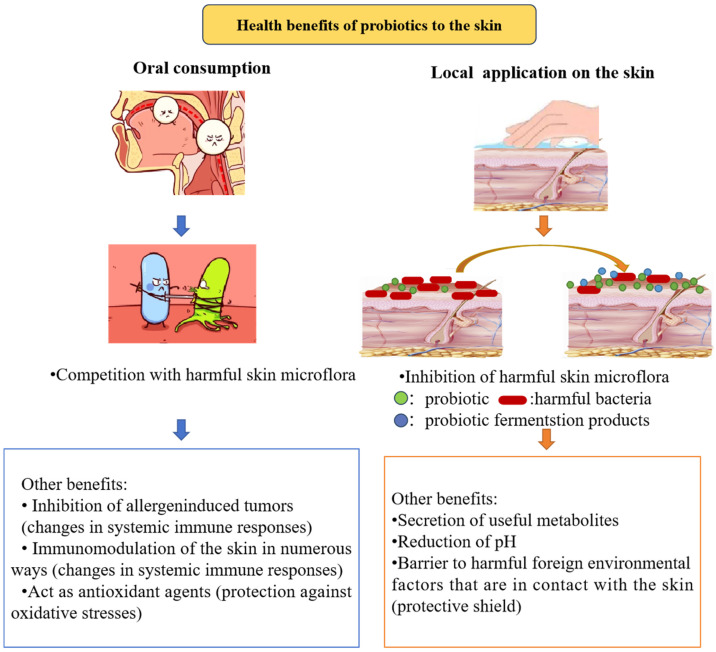
The primary processes by which probiotics promote skin health.

**Figure 2 molecules-28-06765-f002:**
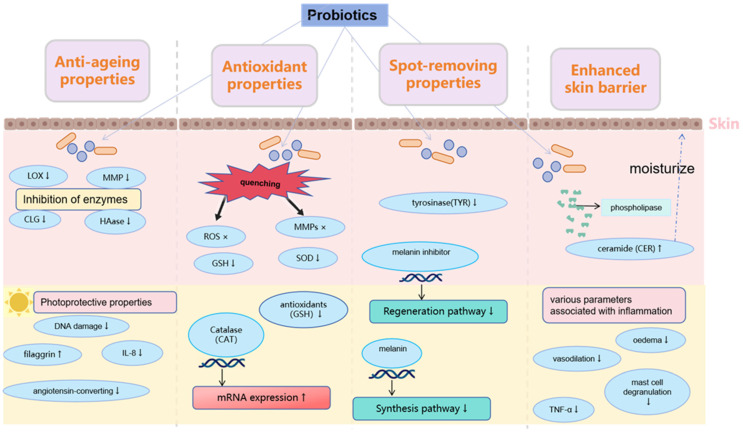
The significant cosmetic properties of probiotics. (↑—Increased expression,↓—Reduced expression)The effect of probiotics on the free radicals is multidirectional and includes the direct quenching of reactive oxygen (ROS) and matrix metalloproteinases (MMPS) species; the enhancement of endogenic antioxidant enzyme production (SOD (superoxide dismutase) and glutathione); the inhibition of enzymes involved in ROS generation (glutathione S-transferase, microsomal monooxygenase, mitochondrial succinoxidase, or NADH oxidase); the protection and regeneration of antioxidant compounds (vitamin C or E).

**Figure 3 molecules-28-06765-f003:**
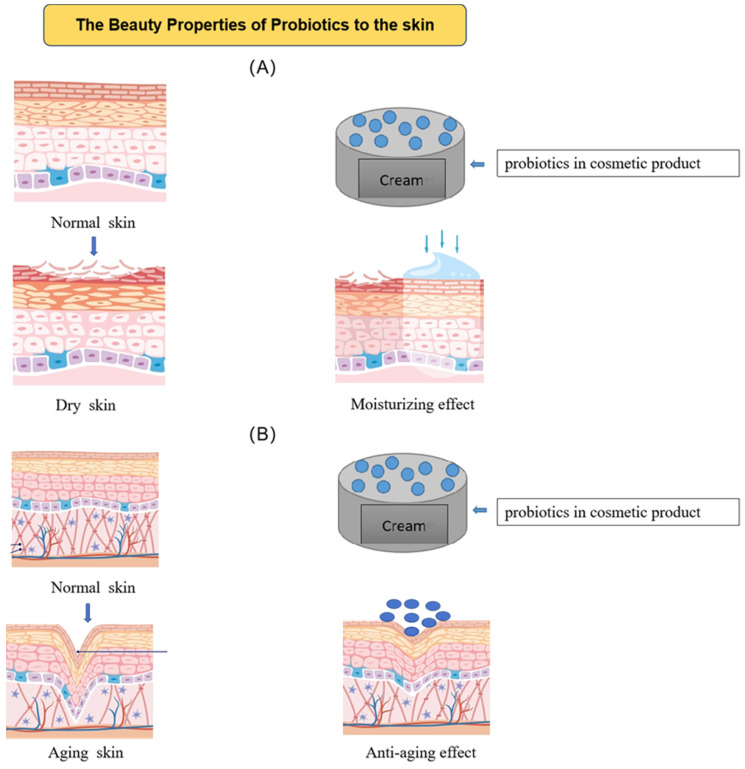
Anti-aging and moisturizing effects of probiotics. (**A**) Probiotics improve skin hydration; (**B**) probiotics reduce skin wrinkles.

**Figure 4 molecules-28-06765-f004:**
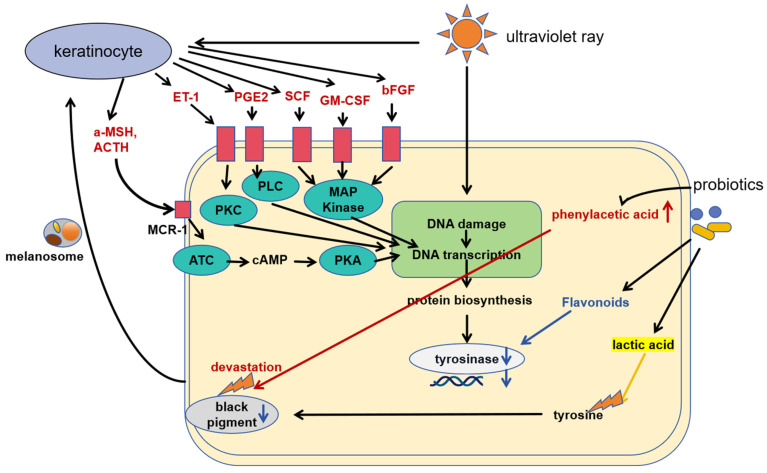
Mechanism of action of probiotics to inhibit melanogenesis. Probiotics inhibit tyrosinase activity by producing tyrosinase inhibitors; probiotics inhibit tyrosinase activity by promoting the production of phenylacetic acid; probiotics destroy tyrosine by producing lactic acid; probiotics destroy melanin by producing substances that degrade melanin.

**Figure 5 molecules-28-06765-f005:**
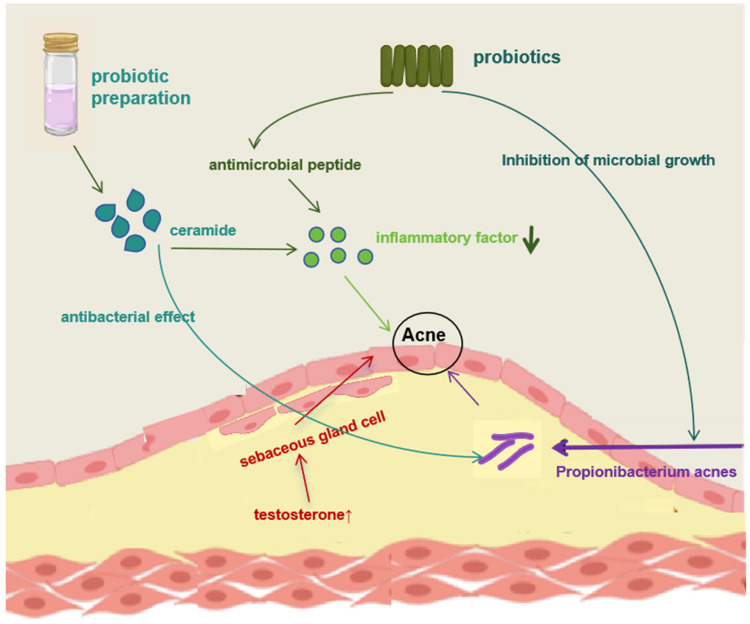
The role of probiotics in the treatment of acne. The main causes of acne are increased secretion of androgens, which contributes to an increased rate of sebaceous cell production; rapid multiplication of propionibacterium acnes; inflammatory reactions.

**Figure 6 molecules-28-06765-f006:**
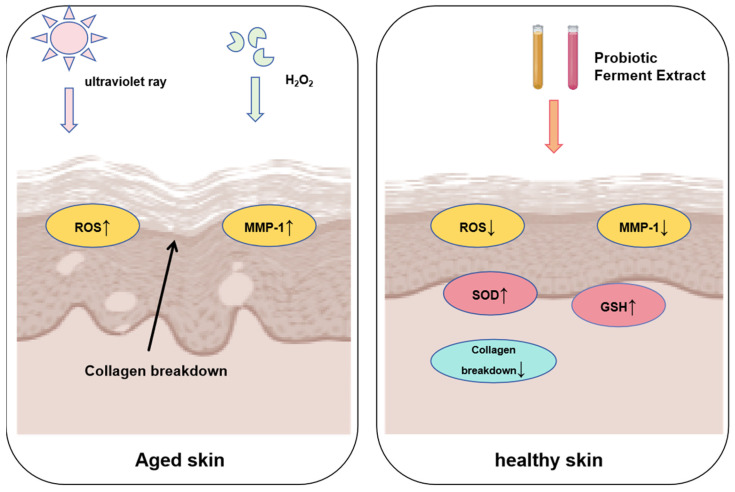
Role of probiotic fermentation in anti-aging. (↑—Increased expression, ↓—Reduced expression).

**Table 1 molecules-28-06765-t001:** Selected examples of cosmetic products containing probiotics. * The ingredient listed as “Bifida ferment lysate” corresponds to a lysate from *Bifidobacterium longum* reuter.

Product ID	Cosmetic Effects (Manufacturer′s Declaration)	Type of Product	Cosmetic Ingredient List
1	Skin feels soft, moisturized, and revitalized. Restore skin barrier function and seal in moisture on the skin’s surface. Skin relief and a reduction in potential irritation.	Lotion	*Lactobacillus* ferment
2	Enhance wrinkles, unwind, and keep skin looking young. Enhance wrinkles, unwind, and keep skin looking young.	Lotion	*Lactococcus* ferment extract Yeast cytolytic extract
3	To stop dry air from the outside, stop water loss and create a water-locking barrier on the skin’s surface.	Lotion	*Lactobacillus* ferment hyaluronic acid
4	Maintain the balance between the skin’s water and oil content. Improved skin stability. Revitalize and repair skin.	Toner	Bifida ferment lysate *
5	Multidimensional repair of skin fragility and relieving skin discomfort.	Cream	*Alteromonas Baumann* ferment extract
6	Adjust skin and facial flora, inhibit acne inflammation root, improve redness and sensitivity, and enhance skin defense.	Cream	*Lactobacillus* extract
7	Balance skin micro-ecological environment and repair the micro-ecological barrier. Intensive moisture lock water and reduce skin moisture loss.	Cream	*Leuconostoc* ferment filtrate Prebiotics: pentavitin
8	Balance skin flora and improve skin health.	Cream	*Lactobacillus*/soymilk ferment filtrate
9	Balance skin flora and improve skin health.	Cleanser	*Lactobacillus*/soymilk ferment filtrate Candida bobicola/glucose/methyl Rapeseedate ferment
10	Improve skin antioxidant capacity.	Cream	*Lactococcus* ferment Bacillus ferment
11	Produce acne suppressor, help regulate the skin surface flora, and reduce the risk of acne.	Cream	Bifidobacterium longum, lysate
12	Repair skin barrier, and relieve redness and discomfort.	Mask	*Vitreoscilla* ferment
13	Promotes collagen regeneration and smoothes wrinkles.	Lotion	Bifidobacterium longum, lysate Yeast extract *Lactococcus* ferment
14	Reduce skin redness and sensitivity. Inhibiting skin oxidation factor.	Serum	PITERA^TM^
15	Regulate skin flora, strengthen the micro-ecological barrier, and improve skin condition.	Lotion	*Lactobacillus*/soymilk ferment filtrate Bifidobacterium longum, lysate
16	Strengthen the skin barrier and improve the skin quality. Strengthen the muscle base, firm the skin, and delicate skin. Promote collagen regeneration.	Serum	Bifidobacterium longum, lysate Yeast extract *Vitreoscilla* ferment
17	Repair the skin barrier.	Serum	Bifidobacterium longum, lysate Yeast extract *Lactobacillus*
18	Maintain skin micro-ecological balance and adjust skin flora.	Serum	*Lactobacillus*/soymilk ferment filtrate
19	Prevent the imbalance of micro-ecological barriers. Balance bacteria symbiosis, improve skin redness.	Serum	*Lactobacillus* ferment lysate
20	Enhance the outer strength of the skin.	Mask	*Lactobacillus*
21	After fermentation, lactic acid bacteria present a protective film to prevent water evaporation, water tender, and shining through the skin, and strengthen the skin cuticle.	Cream	*Lactococcus* ferment lysate Bifida ferment extract
22	Activate skin micro-ecological activity and inhibit harmful bacteria reproduction, and the skin becomes soft and delicate.	Lotion	*Lactobacillus*
23	Balance the micro-ecological barrier to help reduce repeated breakouts. Balance water and oil, stabilize skin, and maintain skin health.	Gel Cream Toner	*Lactobacillus* *Lactobacillus* ferment *Lactococcus* ferment lysate Prebiotics: chicory root extract
24	Make the skin moist and tender. Maintain skin elasticity and permeability.	Cream Cleansing	Original nourishing ingredient: S.E. essence Bifidobacterium/soybean ferment *Lactococcus*/milk ferment
25	Balance skin micro-ecology and make skin healthy and delicate.	Serum Cream	NATURAL RE^TM^: *Lactobacillus* complex extract
26	Stong microbial barrier. Maintain the balance of bacteria and stabilize the healthy state of the skin.	Cleansing Toner	*Lactobacillus* *Lactococcus* ferment lysate *Lactobacillus*/soymilk ferment filtrate Saccharomyces/rice ferment filtrate
27	Prebiotics: maintain skin micro-ecological health. Biostime: provides an environment for probiotics to grow.	Serum Cream	Inulin Alpha-glucan oligosaccharide Yeast ferment extract filtrate *Lactobacillus*/soymilk ferment lysate *Lactobacillus*/soybean extract ferment filtrate. *Lactobacillus*/punica granatum feuit ferment extract
28	Extracted from the fermentation of lactic acid bacteria, it helps to reduce the effects of toxins and maintain the balance and integrity of the skin microbiome.	Serum	*Lactococcus* ferment extract
29	Firming skin and energizing bacteria.	Serum	Plant-derived probiotics: chicory root
30	Reduce skin redness, acne, and sunburn. Restore skin elasticity.	Serum	*Lactococcus lactis* fermentation lysate
31	Maintain and counterbalance facial symbiotic bacteria. Help restore the dynamic balance of bacteria on the skin surface and create a stable skin microenvironment.	Cleaning Serum Cream	*Lactobacillus/soymilk* ferment filtrate *Lactobacillus*
32	Balance skin flora and prevent aging.	Serum	Bifida ferment lysate
33	Regulate skin pH.	Cleaning	Bifidobacterium longum, lysate *Lactobacillus* ferment *Lactobacillus* ferment lysate *Streptococcus thermophilus* ferment *Lactobacillus*/soybean extract ferment
34	Maintain the balance of bacteria and stabilize the healthy state of the skin.	Serum	Bifida ferment lysate
35	Accelerate collagen regeneration and effectively improve skin fullness.	Cream	Bifidobacterium longum, lysate

**Table 2 molecules-28-06765-t002:** Experimental in vitro and in vivo studies of the beneficial effects of probiotics and their active ingredients on the skin and their mechanisms of action-summary.

Activity	Experimental Model		Type of Probiotics /Active Constituent	Mechanism of Action/Effect	References
	In Vitro	In Vivo/Ex Vivo			
Antioxidant	Human dermal fibroblast (HDF)		Lipoteichoic acid isolated from *Lactobacillus plantarum*(LTA) - LTA pretreatment	- Inhibited MMP-1 expression. - Inhibited activation of extracellular signal-regulated kinases (ERK) and c-Jun N-terminal kinases (JNK). - Promoted type 1 procollagen synthesis and reduced the generation of ROS induced by UV irradiation.	[14]
	Normal human dermal fibroblast (NHDF) cells B16F10 murine melanoma cells		Heat-killed *Lacticaseibacillus* *paracasei* (PL)	- Reduced DNA damage. - Alleviated UVB-induced oxidative damage. - Attenuated UVB-induced photoaging.	[15]
		Murine	Nicotinamide mononucleotide (NMN) combined with *Lactobacillus fermentum* TKSN041	- Improved murine skin damage caused by UVB irradiation and the protective mechanism. - Increased the protein expression levels of AMPK, IκB-α, SOD1, and CAT in the skin tissues and Ireduced protein expression of NF-κBp65.	[16]
	Mouse skin fibroblast (MSF) cells Human epidermal melanocytes (HEM)		Heat-killed *L. rhamnosus* ATCC 7469 (RL)	- Absorbed UVB and reduced DNA damage. - Downregulated MMP-1, 2, 3 expressions associated with MAPK signaling. - Reduced ROS content. - Suppressed tyrosinase and TYRP-2 activity and/or levels associating with PKA/CREB/MITF signaling.	[17]
	Human keratinocytes Human dermal fibroblasts B16F10 murine melanoma cells		Tyndallized *Lactobacillus acidophilus* KCCM12625P (AL)	- Induced anti-wrinkle effects by regulating wrinkle-related genes. - Reduced the mRNA expression of melanogenesis-related genes such as tyrosinase, TYRP-1, and TYRP-2.	[18]
Anti-aging	HS68 cells dermal fibroblast cells		Extracts of *Jasminum sambac* flowers fermented by *Lactobacillus rhamnosus*	- Enhance the viability of HS68 cells. - Remarkably attenuate the UVB/H_2_O_2_-induced excessive production of reactive oxygen species, degradation of collagen, and premature senescence. - Enhance the expression of antioxidant genes.	[19]
			*Streptococcus salivarium* spp. *Streptococcus thermophilus* S244	Significant increase in skin moisture (immediate and long-term).	[20]
	Hs68 cells Human dermal fibroblasts	Hairless mice	Administered vehicle or *L. plantarum* HY7714 - Oral supplementation (1 × 10^9^ CFU, per day) for 8 weeks	- Significant increase in ceramide level flow compared to the UVB group. - Suppressed the increased transepidermal water loss and decrease in skin hydration. - Improved the reduction in SPT mRNA levels and suppressed the increase in ceramidase mRNA levels caused by UVB. - Effectively rescued UVB-reduced procollagen expression through the inhibition of UVB-induced matrix metalloproteinase expression in human dermal fibroblasts. - Inhibited the number, depth, and area of wrinkles in hairless mouse skin.	[21,22]
	Human foreskin fibroblast (Hs68)	SKH-1 hairless mice	Fermented blackberry (FBB) by *L. plantarum* JBMI F5 -FBB pretreatment -FBB administration	- Inhibited UVB-mediated type-1 procollagen degradation and (MMP)-1 and MMP-2 protein expression. - Suppressed NF-κB. - Activation and MAPK phosphorylation. - Diminished the wrinkle formation in dorsal skin and epidermal thickening in UVB-irradiated hairless mice.	[23]
	HaCaT cells	SKH-1 hairless mice Volunteers	Kimchi-derived *L. plantarum* K8 lysates - Oral supplementation for 8 weeks - Oral supplementation experimental candy containing 2.1% *L. plantarum* K8 lysate	- Increased hyaluronic acid content - Decreased epidermal thickening. - Reduced damage to barrier function. - Significant increase in hydration. - Decreases in horny layer thickness and TEWL value were observed on the face and forearm.	[24,25]
		Double-blind, placebo-controlled trial Japanese women volunteers(aged 31–62 years) 8-week treatment	Heat-killed cells of *Lactococcus lactis* strain H61 - Oral supplementation (60 mg per day) for 8 weeks	- Decreased skin elasticity and melanin Content in the cheek. - Increased sebum content. - Apparent hair follicles and dryness of the throat at week 8 were higher in the overall H61 group than in the combined placebo group. - Marked improvements in self-surveyed skin elasticity.	[24,25]
	UVB-irradiated normal human epidermal keratinocytes (NHEKs)		Cosmetic preparation that contained Water extract from heat-killed *L.lactis* H61	- Suppression of inflammation of the skin. - Absorbed electromagnetic radiation in the UVB range. - Inhibited the production of interleukin-8 induced by UVB. - Did not protect against hydrogen peroxide-induced cell damage.	[26]
	Primary epidermal cells	Hairless mice	Heat-killed *L. plantarum* L-137	- Suppressed the loss of water content in the stratum corneum. - Increased HA production.	[27]
		Clinical trials	Extracellular vesicles (EVs) that were secreted from *L. plantarum* of women in their 20s (LpEVs)	- Suppressed wrinkle formation and pigmentation.	[28]
Spot removing and whitening			Cultures of *Bifidobacterium bifidum* IDCC4201 and *Lactiplantibacillus plantarum* IDCC 3501 - Phenyllactic	- Anti-tyrosinase. - Reduced melanin synthesis. - Altered protein expression associated with the melanogenesis pathway.	[29]
			Kimchi-derived *Pediococcus acidilactici* PMC48	- Degrades melanin. - Tyrosinase-inhibiting effect.	[30]
	B16F10 mouse melanoma cells		Lipoteichoic acid (LTA) isolated from *Lactobacillus plantarum* (pLTA)	- Inhibited melanogenesis. - Reduced the cellular activity of tyrosinase and the expression of tyrosinase family members in a dose-dependent manner. - Reduced the expression of microphthalmia-associated transcription factor (MITF).	[31]
		Clinical observation	*L. plantarum*-GMNL6 - The external ointment	- Enhanced collagen synthesis and the gene expression of serine palmitoyltransferase small subunit A. - Reduced melanin synthesis, the biofilm of *Staphylococcus aureus*, and the proliferation of cutibacterium acnes. - The syndromes of skin moisture, skin color, spots, wrinkles, UV spots, and porphyrins were improved.	[32]
	Anti-melanogenic signaling pathway in α-melanocyte stimulating hormone (α-MSH)-treated B16F10 melanoma cells and zebrafish		Extracts of Rhodobacter sphaeroides (Lycogen™)	- Inhibited the cellular melanin contents and expression of the melanogenesis-related protein, including microphthalmia-associated transcription factor (MITF) and tyrosinase. - Reduced phosphorylation of MEK/ERK without affecting phosphorylation of p38. - Decreased zebrafish melanin expression in a dose-dependent manner.	[33]
Anti- inflammatory		Twenty-seven AD patients and six healthy control subjects Staphylococcus aureus-induced mouse AD models	*Lactobacillus plantarum*-derived extracellular vesicles - Administration	- Reduced epidermal thickening and the IL-4 level. - Treatment prior to *S. aureus* EV treatment.	[34]
		Ex vivo skin models	Live and the lysate products of probiotic strain *Lactobacillus reuteri* DSM 17938	- Reduced proinflammatory IL-6 and IL-8. - Live. - Increased aquaporin 3 (AQP3) gene expression. - Had antimicrobial action against path-genic skin bacteria (staphylococcus aureus, streptococcus pyogenes M1, cutibacterium acnes AS12, pseudomonas aeruginosa). - The lysate-enhanced laminin A/B levels in a healthy (unstimulated) state of RHE.	[35]
	Human keratinocytes	In the forearm skin of 11 atopic dermatitis (AD) patients stratum corneum 20 healthy elderly women.	An experimental cream containing sonicated *Streptococcus thermophilus* - 2-week application of the cream 7d application of the cream	- Increase in skin ceramide amounts. - Improvement of the signs and symptoms characteristic of AD skin. - Increase in skin ceramide amounts. - Improvement of lipid barrier and more effective resistance against xerosis.	[36]
	Nerve cell cultures in vitro	Ex vivo human skin explant model Sixty-six female volunteers	*Bifidobacterium longum* sp. extract (BL) - Either the cream with the bacterial extract at 10% (the face, arms, and legs twice a day for two months)	- Significant improvement versus the placebo in variousparameters associated with inflammation. - Inhibited capsaicin-induced CGRP release by neurons. - Decrease in skin sensitivity at the end of the treatment. - Increase skin resistance. - Physical and chemical aggression compared to the group of volunteers increased skin resistance.	[37]

## Data Availability

Data will be made available upon request.

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
