# Peer review of "Applications of Probiotic Constituents in Cosmetics"

_molecules, 2023, doi:10.3390/molecules28196765_

Round 1
Reviewer 1 Report
The topic is very interesting and well written. It contains extensive information pharmaceutical applications of Probiotics Constituents in Cosmetics. However, the novelty is very limited as many reviews recently discuss the same idea. I don’t recommend publication unless the authors show novelty over those similar reviews. The review is well organized. The conclusion and the abstract should be rewritten in a more precise and informative way.
1. In abstract, Provide changes in the abstract starting with the submission of the manuscript. What is the review about? How will it be addressed? what gaps do the authors want to fill with this manuscript!!!
2. What novelty over old published review articles in this area should highlighted in the abstract and the introduction too.
https://onlinelibrary.wiley.com/doi/full/10.1111/jocd.14734
Application and mechanism of probiotics in skin care: A review
https://www.mdpi.com/1420-3049/26/5/1249
Probiotics in Cosmetic and Personal Care Products: Trends and Challenges
Although this article was cited but it should be compared in terms of novelty. What is novelty over this article?
https://www.mdpi.com/2072-6643/15/14/3123
The Role of Probiotics in Skin Health and Related Gut–Skin Axis: A Review
https://link.springer.com/article/10.1007/s00253-022-12116-5
Current postbiotics in the cosmetic market—an update and development opportunities
https://www.hindawi.com/journals/bmri/2023/2375223/
Microbial Biosurfactant as an Alternate to Chemical Surfactants for Application in Cosmetics Industries in Personal and Skin Care Products: A Critical Review
3. The aforementioned articles should be cited and mentioned in the introduction as old similar reviews.
4. In line 35 of introduction, what are the two categories exactly ? Probiotics and epigenetics are two product categories
5. In line 45, cfu/g full name should be mentioned in the first mentioning [ colony-forming unit per gram.]
6. IN LINE 56, it is FAO Food and Agriculture Organization OR FDA , CORRECT IT PLEASE
7. TABLE 1 SHOULD BE NUMBERED FOR THE PRODUCTS 1,2,3,…. Additionally, what is new over table 2 in this article https://www.mdpi.com/1420-3049/26/5/1249
8. Line 94, mention full name for ROS
9. Title for table 2 should be stated in more details to reflect the various discussed parameters.
10. Line 220-221 needs appropriate reference
11. 3.1, 4.2, and 5 could be represented as figure. Generally, more figures should be provided.
12. UVB short definition is required in line 284. Also ,UVA line 379.
13. Lines 407 till 419 should be moved to the introduction
14. Conclusion should be more precise and illustrate new findings about the title.
15. Abbreviation list is strongly recommended
16. More illustrative diagrams could improve the quality of this review. More figures and less text strategy will improve the overall presentation for this review.
17. Future research plans should be highlighted
Best wishes
Reviewer 2 Report
Some concepts should be corrected in the manuscript. Epigenetics are environmental factors that influence how genes are expressed and work. They are not product category (line 35)!!! Also, apigenin is a flavone but not an added active product of probiotics (Lines 76-79)! In addition, UVR irradiation (line 157) should be corrected into UV radiation.
The manuscript should undergo extensive editing of English language.
Reviewer 3 Report
As requested, I’ve reviewed the manuscript titled "Applications of Probiotics Constituents in Cosmetics" and I found it to be of much interest to the readers. This review covers all the major aspects of the theme – introduction covering information about probiotics and their application in cosmetic products, application of probiotics in anti-aging cosmetics, function of the products and their ingredients. Also, the manuscript is easy to read. However, some minor issues should be corrected: 1. Table 1 indicates list of ingredients, but the information provided is about ingredients produced from LAB strains, not the whole list of the products included in a cosmetic product. It would be appropriate to change the name and find a more appropriate one. 2. Figure 3 should be placed somewhere in the text before the conclusion part and not in it. Also, information provided about Fig. 3 should be provided in other part of the manuscript, not in the conclusion.The manuscript seems to be of good quality, so not much can be added to the review report.
Round 2
Reviewer 1 Report
the authors did all required changes. the paper could be published in the current form. Greetings